# Flexible Stretchable, Dry-Resistant MXene Nanocomposite Conductive Hydrogel for Human Motion Monitoring

**DOI:** 10.3390/polym15020250

**Published:** 2023-01-04

**Authors:** Yafei Liu, Huixia Feng, Yujie Gui, Ting Chen, Haidong Xu, Xiaoxue Huang, Xuemei Ma

**Affiliations:** 1School of Petroleum and Chemical Engineering, Lanzhou University of Technology, Lanzhou 730050, China; 2School of Chemistry and Chemical Engineering, Normal College for Nationalities, Qinghai Normal University, Xining 810008, China

**Keywords:** MXene-based hydrogel, temperature tolerance, mechanical flexibility, anti-drying, strain sensors

## Abstract

Conductive hydrogels with high electrical conductivity, ductility, and anti-dryness have promising applications in flexible wearable electronics. However, its potential applications in such a developing field are severely hampered by its extremely poor adaptability to cold or hot environmental conditions. In this research, an “organic solvent/water” composite conductive hydrogel is developed by introducing a binary organic solvent of EG/H_2_O into the system using a simple one-pot free radical polymerization method to create Ti_3_C_2_T_X_ MXene nanosheet-reinforced polyvinyl alcohol/polyacrylamide covalently networked nanocomposite hydrogels (PAEM) with excellent flexibility and mechanical properties. The optimized PAEM contains 0.3 wt% MXene has excellent mechanical performance (tensile elongation of ~1033%) and an improved modulus of elasticity (0.14 MPa), a stable temperature tolerance from −50 to 40 °C, and a high gauge factor of 10.95 with a long storage period and response time of 110 ms. Additionally, it is worth noting that the elongation at break at −40 °C was maintained at around 50% of room temperature. This research will contribute to the development of flexible sensors for human-computer interaction, electronic skin, and human health monitoring.

## 1. Introduction

Conductive hydrogel has been ubiquitously used in a wide range of wearable devices [1,2,3,4,5,6], soft robots [7,8,9,10], tissue engineering [11,12,13], wearable electronics [14,15,16], artificial skin [17,18,19], health monitoring [20,21,22], smart medical therapy [23,24,25,26] and supercapacitors [27,28,29]. Although conventional hydrogels’ mechanical and electrical properties are affected by the evaporation of water, they cannot sustain their mechanical and electrical conductivity properties over time, which severely restricts their practical applications. Therefore, research into the long-term stability and reliable moisture retention of conductive hydrogels for flexible sensors is essential.

Currently, the introduction of organic solvents into hydrogel matrices is an effective method for preparing freeze- and dry-resistant organic conductive hydrogels [30,31,32,33]. For example, a water-glycerol binary solvent was used by Li et al. [16] to create an organic conductive hydrogel that not only has superior mechanical characteristics (1000% tensile elongation) and improved electrical conductivity (1.34 S/m) but can also function appropriately between −20 and 80 °C. However, conventional hydrogels are deficient in a conductive medium, such as polypyrrole [34,35,36] and carbon materials [37,38,39], which prevents them from creating a fully functional conductive network, resulting in low conductivity and severely restricting the range of application scenarios. The MXene sheets have attracted much interest in various applications due to their high aspect ratio morphology, excellent solution processability, and rich surface chemistry. In recent research conducted, it has been discovered that MXene-based hydrogels possess different intriguing and valuable qualities, such as photoredox catalysis, photothermal behavior, and sense [40]. However, the exceptional conductivity of the resultant hydrogels can be achieved by assembling MXene sheets into polymer architectures due to the excellent metallic conductivity of the MXene sheet. For example, Liao et al. [41] developed a MXene nanocomposite organic hydrogel using the solvent replacement method that can be prepared as a freeze-resistant, self-healing strain sensor with a wide strain detection range (up to 350%) and high sensitivity (GF = 44.85).

Furthermore, polyvinyl alcohol/polyacrylamide/MXene hydrogel (PAEM) containing water-glycerol hybrid solvents prepared by free radical polymerization in a single pot was observed. The use of MXene nanosheets creates an efficient conductive channel for PAEM. Furthermore, the MXene-based hydrogel exhibits good structural stability and reliability across a broad temperature range of −50 to 40 °C. Additionally, by simply adjusting the glycerol/water ratio, the prepared hydrogels can maintain their conductivity at 79.9% of their initial conductivity and their weight at over 95% of their initial weight after 7 days at room temperature. The optimized PAEM hydrogels have excellent mechanical properties (1033% strain) and high sensitivity (GF = 10.95). Furthermore, PAEM can be incorporated into pressure sensors. This research offers a practical answer for the use of smart electronic skins in harsh environments.

## 2. Materials and Methods

The Ti_3_AlC_2_, polyvinyl alcohol (PVA, alcoholysis, 99.0–99.4 mol%), and lithium fluoride (LiF, AR) were obtained from the Shanghai Maclean Biochemical Technology Co., Ltd., (Shanghai, China), while ethylene glycol (EG, AR) and N, N′-methylene-bis-acrylamide (MBAA, AR) were obtained from Sinopharm Chemical Reagent Co., Ltd., (Xi’an, China). The acrylamide (AAm, AR) and ammonium persulfate (APS, AR) were also supplied by Shanghai Aladdin Biochemical Technology Co., Ltd., (Shanghai, China). Finally, hydrogen chloride (HCl, AR) was supplied from Tianjin Damao Chemical Reagent Co., Ltd., (Tianjin, China).

The chemical makeup of MPAE hydrogels was investigated using an FTIR spectrum (FTIR-850, Tianjin Gangdong Science and Technology Development Co., Ltd.) (Tianjin, China). The morphology of MXene and hydrogel was observed using scanning electron microscopy (SEM, MIRA LMS, and TESCAN) (after freeze-drying). The structure of MXene nanosheets was examined using XRD (Advance, Bruker D8). The X-ray photoelectron spectroscopy analyzed the material’s constituent elements (Thermo K-alpha, Japan). Additionally, an electrochemical workstation recorded the electrical signals from MPAE strain sensors (CHI660E, Shanghai Chenhua Instruments Co., Ltd.) (Shanghai, China).

The conductive hydrogel samples of various volumes were tested for energy storage modulus (G’) and loss modulus (G″) over the temperature range of −40 to 40 °C using a rheometer (MARS60, Thermo HAAKE, Germany). The sample was shaped like a cylinder with a 20 mm diameter and a 2.3 mm height. The constant strain (γ) was set at 0.1%, and the scanning frequency (ω) was 10 rad/s.

A mass of 0.6 g LiF was added to 20 mL of 9M hydrochloric acid and stirred for 5 min before adding 0.5 g Ti_3_AlC_2_ to the above etching solution and stirring at 40 °C for 48 h, after which the acidic suspension was washed with deionized water until the pH of the supernatant reached 6. At that point, the mixed solution was sonicated for 20 min under N_2_. Finally, the dispersion was centrifuged at 3500 rpm, and the precipitate was collected and vacuum-dried to yield Ti_3_C_2_T_X_ powder (MXene).

Furthermore, a mass of 2 g of PVA powder was mixed with 18 g of deionized water for 2 h at 90 °C with vigorous stirring to form a 10% PVA solution. Following that, deionized water was mixed with 0.02 g MXene, 1.2 g AAm powder, 3 g PVA solution, 4 mL EG solution, 200 μL MBAA solution (1 wt%), and 200 μL APS initiator solution (10 wt%). Subsequently, an in-situ polymerization was performed at 60 °C to produce the PAAm network and hydrogel. The hydrogel formed with PVA and AAM was labeled PA; however, that formed with PVA, AAM, and EG was labeled PAE; while that formed with PVA, AAM, EG, and MXene was named PAEM. The final mass ratio of PVA/AAM was 1:3. The EG content was 0; 8.3; 16.6; 25.0; 33.3; and 41.6 wt %. Additionally, content was labeled PAEM0%, PAEM_8.3%_, PAEM_16.6%_, PAEM_25.0%_, PAEM_33.3%_, and PAEM_41.6%_, respectively. The MXene content was 0.075, 0.15, 0.23, 0.30, and 0.37 wt%. The contents were labeled PAEM_0.075%_, PAEM_0.15%_, PAEM_0.23%_, PAEM_0.30%_, and PAEM_0.37%_, respectively.

## 3. Results

### 3.1. Preparation and Characterization of MXene Nanomaterials

According to Figure 1, the MXene was created by etching and delaminating Ti_3_AlC_2_ precursors with a LiF/HCl solution. MXene material is a two-dimensional, layered structure of metal carbides and metal nitrides collectively. The MXene used in this paper is mainly Ti_3_C_2_T_X_. The morphology of Ti_3_AlC_2_ precursor material is a stacked-layer structure with a relatively large diameter [3], and the morphology of Ti_3_C_2_T_X_ after HF etching would show an accordion shape (Figure 2a,b) because the Al layer in Ti_3_AlC_2_ was removed by HF etching, which leads to the expansion of the layer spacing.

In Figure 2c, which indicates the X-ray diffraction pattern of Ti_3_AlC_2_ material, strong reflection peaks of 9.5° (002), 19.2° (004), 34.1° (101), 38.8° (008), 39.0° (104), and 41.8° (105) can be observed [42]. However, following HF etching, the etching of the Al layer and the embedding of the lithium-ion indicate the weakening of the characteristic peak of the Al element at 39.0° and the shift of the (002) to a smaller angle, demonstrating the successful preparation of MXene. The XPS spectra of the Ti_3_C_2_T_X_ material showed the presence of Ti, C, O, and F elements, as shown in Figure 2d, and the Ti content was 0.14%, C was 53.03%, O was 35.37%, and F was 11.46%, according to the XPS spectral test results.

### 3.2. Design, Synthesis, and Structural Characterization of the PAEM Hydrogels

The AAm, PVA, and MXene were fully dissolved in deionized water, as shown in Figure 3. and then MBAA and APS were added as crosslinker and initiator, and subsequently EG solution was added to the dispersed solution and polymerized in situ at 60 °C. After the initial gel formation, it was placed at −20 °C to further cross-link between PVA and EG molecules, which can encourage the formation of PVA crystal structure domains by forming numerous hydrogen bonds with PVA chains. Additionally, the PVA crystal structure domain can act as a strong bond, effectively improving the organic hydrogel’s mechanical properties. As seen in Appendix A, the MXene nanosheets were uniformly distributed in the hydrogel matrix, which can improve the sensing performance. More importantly, EG can lower the freezing point of H_2_O and prevent PAEM hydrogels from freezing at sub-zero temperatures. In addition to having high toughness and good elasticity, the PAEM hydrogels, with the addition of EG, also improve their mechanical properties and conductivity stability in low-temperature environments.

In the PVA and PVA/EG spectra, the stretching vibration absorption peak of O-H is shown at 3420 cm^−1^ (Figure 4), which has a broad peak shape due to the presence of hydrogen bonding conduction. The stretching vibration absorption peak of O-H in-plane deformation and the stretching vibration absorption peak of C-O are shown at 1430 cm^−1^ and 1070 cm^−1^, respectively [43]. However, based on the PVA and PVA/EG spectra, the PVA/EG/PAM and PVA/EG/PAM/MXene spectra showed distinct PAM characteristic peaks, such as the N-H symmetric, antisymmetric stretching vibration absorption peaks of the primary amine group at 3400~3150 cm^−1^, the C-N stretching vibration absorption peak at 1050 cm^−1^, the C=O stretching vibration absorption peak at 1700 cm^−1^, and the C=C stretching vibration absorption peak near 1500 cm^−1^.

### 3.3. Exploration of Variables in PAEM

Dual-network hydrogels with brittle and ductile networks can have good mechanical properties. The brittle network, which acts as a sacrificial bond, dissipates energy during deformation via an internal fracture, whereas the interpenetrating ductile network provides high tensile properties. Additionally, as shown in Figure 5b, with the addition of acrylamide, the modulus of elasticity was effectively increased, but the elongation at break was subsequently reduced. Subsequently to determining the optimal amount of acrylamide, we added different volume fractions of ethylene glycol to the PVA/PAM hydrogels, and as shown in Figure 5c, the addition of more ethylene glycol solution increased the stretchability of the PAE hydrogels; however, as shown in Figure 5d, with the addition of ethylene glycol, its electrical conductivity also decreased. Therefore, when the volume fraction of EG was 33.3% and the mass ratio of PVA to AAm was 1:3, the PAE hydrogels were obtained with an elastic modulus of 1.4 MPa and an elongation at break of 1033% at room temperature, so this condition was used as the sample for further study.

The MXene nanomaterial itself has an ultra-high conductivity (1600 S/m); as a result, the addition of MXene sheet material doped into the PAEM hydrogel can significantly increase the conductivity. It can be seen from Figure 5e that the conductivity of PAEM hydrogel increases with the increase in MXene material content. However, with the addition of an excessive amount of MXene lamellar material, the stacking of nanosheets reduces the strain sensor’s response. Figure 5f shows that the sensitivity of PAEM conductive hydrogel increases and decreases with increasing MXene content, and the sensitivity factor GF is 10.95, reaching a maximum when the mass of MXene material is 0.30%.

### 3.4. Temperature Tolerance

Furthermore, it is difficult to keep hydrogels from drying out and to maintain their long-lasting moisture retention properties. As can be seen from Figure 6a, the water in PAEM hydrogels slowly evaporates over time, and PAEM hydrogels that lose most of their water after long storage have a greater impact on their electrical conductivity and sensing properties. However, by adding EG solution to the MAP hydrogel, its water can be retained to a large extent because strong hydrogen bonds form between EG molecules and H_2_O molecules, which compete with water molecules for hydrogen bonds, preventing the loss of water under room temperature conditions. Additionally, as shown in Figure 6b, we conducted a weight loss test at 30 °C and 35% humidity for 7 days and could see that the drying behavior of PAEM was observed to be influenced by the volume of EG. When the volume fraction of EG added was 33.3%, PAEM could keep more than 95% of its initial weight after 7 days, whereas PAEM 0% could only keep 50% of its initial weight. Compared with the soaking method, the direct addition of EG can improve the anti-drying property of the hydrogel more effectively. Interestingly, the increase in the weight of MPAE occurred with the increase in the volume of added EG, which was due to the absorption of moisture from the air by PAEM.

It can be shown from Figure 6c that the conductivity of PAEM with the addition of EG solution can be well maintained after 7 days of standing, while without the addition of EG solution, the conductivity can only be maintained at 10.8% of the initial conductivity, and when the volume fraction of EG is added at 16.6%, the conductivity can be maintained at 79.9% of the initial conductivity. The PAEM has a long-lasting moisture retention effect because the large amount of EG in it is non-volatile and has a low vapor pressure and moisture absorption. The results show that adding EG solution to PAEM hydrogel can maintain excellent mechanical properties and resistance after long-term storage.

The dynamic thermodynamic analysis (DMA) was used to measure the energy storage modulus (G’) and loss modulus (G″) of PAEM at various temperatures to evaluate the impact of EG content on the freezing resistance of PAEM. One of the most crucial measures of a hydrogel’s mechanical characteristics is its energy storage and loss moduli. The hydrogel’s internal polymer network will be extruded and deformed in response to a specific external force, and the molecular chain segments will be compressed and folded with high potential energy, part of which will be stored by the molecular chain and used to counteract the deformation caused by the external force, which is macroscopically manifested as the rigidity of the material, while the other part of the energy is used for the internal consumption of the molecular chain movement, which is macroscopically manifested as the viscosity of the material. Additionally, as shown in Figure 7a, the rapid increase in G’ and G″ of PAEM without the addition of EG in the temperature range of 10 to −10 °C indicates that icing occurs in this temperature range. The PAEM with the addition of EG, on the other hand, has a lower modulus and can retain strength and elasticity at lower temperatures. As illustrated in Figure 7f, the icing temperature of PAEM hydrogels can be reduced to −18 °C when the volume fraction of EG is 41.6%. However, in the temperature range of −50 to 40 °C, the energy storage modulus of PAEM is consistently higher than the loss modulus, demonstrating the hydrogel’s superior viscoelasticity. The PAEM hydrogel was also tested at room temperature and at −40 °C under extreme conditions, as shown in Appendix A. The elongation at break of the hydrogel in the extreme environment remained at 50% of room temperature, extending the hydrogel’s range of applications even further.

### 3.5. Electromechanical Performance

The change in PAEM resistance is caused by the reversible sliding between the MXene sheet layers of material during the generation of deformation. It is believed that the MXene nanosheets are in contact with each other in the hydrogel matrix, forming a reversible sliding conductive network structure that results in excellent electrical conductivity. As PAEM is subjected to smaller strains, the MXene sheet materials slide, which leads to an increase in the spacing between the MXene sheet materials, at which point the resistance of the hydrogel also increases. Additionally, as the PAEM stretches to a greater extent, the spacing between the MXene lamellae grows wider until they separate from one another, disrupting the formation of the conductive network and resulting in a relatively high resistance.

In addition, the APEM hydrogel is used to test for flexible wearables. As shown in Figure 8a,b, the PAEM-based strain sensor can detect the change in resistance of the finger at bending angles of 60° and 90°. The resistance value of the PAEM sensor decreases, and the relative resistance change rate rises as the finger’s bending angle increases. As PEAM extrudes, the layer spacing of the MXene material decreases, increasing the contact area between the MXene sheet layers of material. A metal-like conductivity emerges, increasing the number of carriers flowing from between the MXene sheets, overcoming the limitation of the polymer network’s limited number of conductive channels, and allowing PAEM to exhibit increased conductivity and an increased number of electron-directed movements. Figure 8c,d shows the resistive signal of our PAEM stretched to 100% with a fast response (110 ms). However, the PAEM-based strain sensor can be connected wirelessly to transmit electrical signals to smart devices such as mobile phones and watches via NFC capability, thus fulfilling the need for real-time monitoring. The results above demonstrate that the PAEM strain sensor is a susceptible sensor for monitoring human motion information and can accurately detect small human motion information, such as finger flexion.

## 4. Discussion

In conclusion, by incorporating MXene into a polymer network composed of a water-glycerol binary solvent system, a temperature-tolerant, electrically conductive, and mechanically robust nanocomposite hydrogel was created. Adding a low content of MXene sheets significantly improved the hydrogel’s electrical performance. Furthermore, because of the formation of strong hydrogen bonds between water molecules and glycerol, the formation of ice crystal lattices (at low temperature) and water evaporation (at high temperature) were effectively restrained, resulting in temperature-tolerant performance (from −50 to 40 °C). The optimized hydrogel contains 0.3 weight percent MXene not only has excellent mechanical properties (a tensile elongation of 1033%) and an improved modulus of elasticity (0.14 MPa), but it also has a high gauge factor of 10.95 with a long storage period and response time (110 ms). As a result, the strategy described in this paper could pave the way for the development of versatile hydrogel materials, which are promising materials for electronic devices such as personalized healthcare.

## 5. Patents

An MPAE conductive composite hydrogel and its preparation method and application (Patent No. CN202211007829.5).

## Figures and Tables

**Figure 1 polymers-15-00250-f001:**
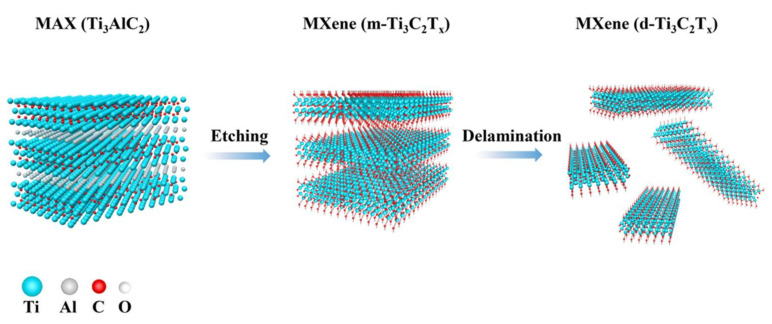
Schematic diagram of the preparation of MXene nanosheets.

**Figure 2 polymers-15-00250-f002:**
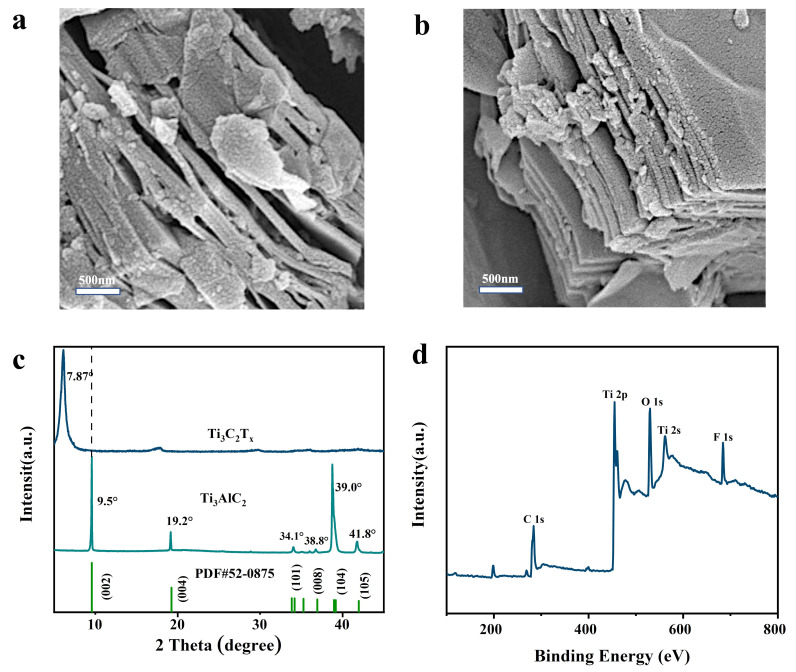
(**a**,**b**) SEM image of MXene material; (**c**) XRD spectra of Ti_3_AlC_2_ and Ti_3_C_2_T_X_ nanosheets; (**d**) XPS spectra of Ti_3_C_2_T_X_ nanosheets.

**Figure 3 polymers-15-00250-f003:**
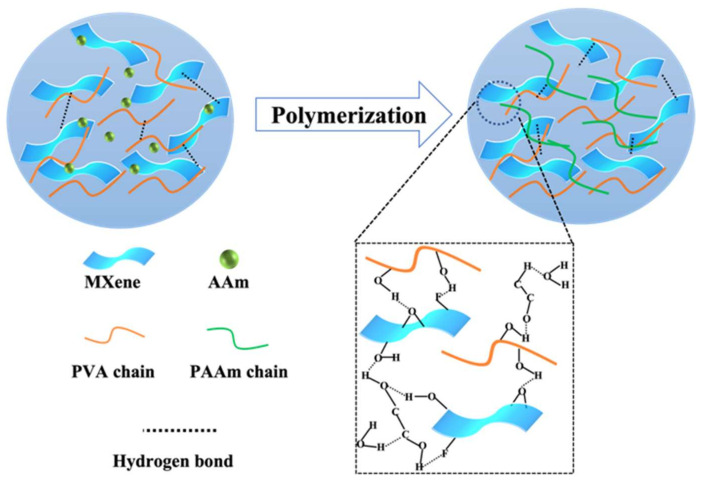
Schematic illustration of the fabrication and interaction of PAEM hydrogels and their gel process.

**Figure 4 polymers-15-00250-f004:**
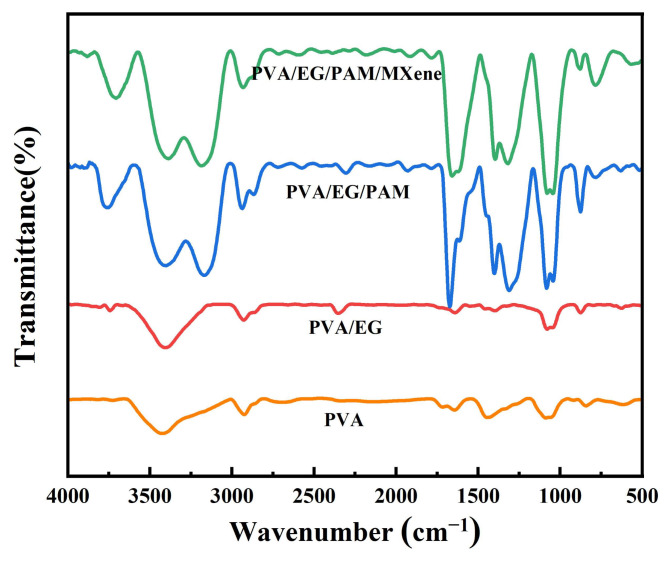
FTIR spectra of PVA hydrogel, PVA/EG hydrogel, PVA/EG/PAM hydrogel, and PVA/EG/PAM/MXene hydrogel.

**Figure 5 polymers-15-00250-f005:**
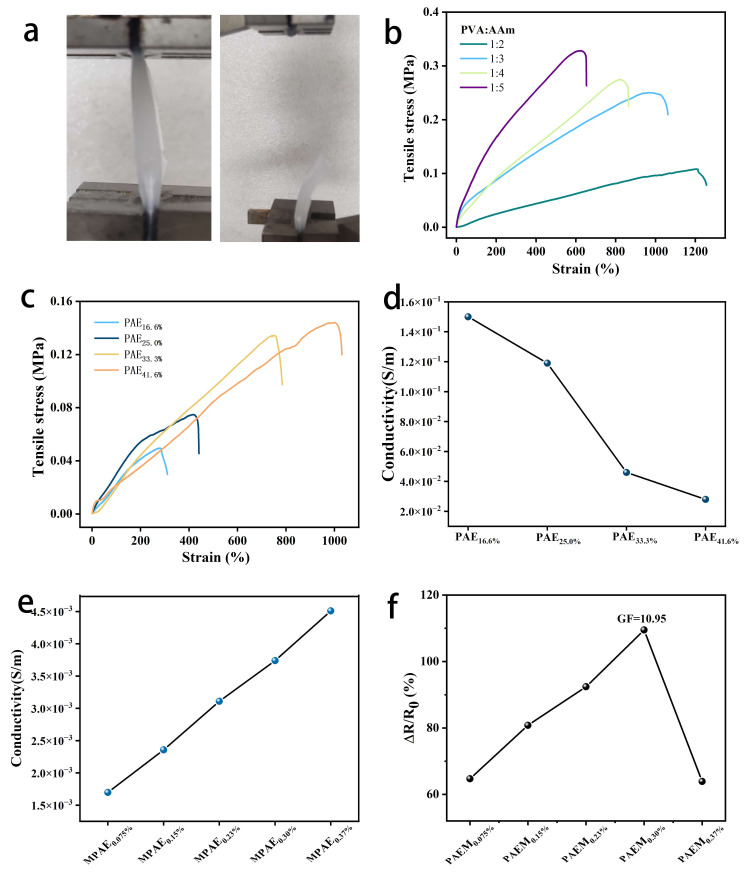
(**a**) Tensile and fracture of hydrogels; (**b**) Tensile stress-strain curves of PA hydrogels (PVA:AAM = 1:2, 1:3, 1:4, and 1:5); (**c**) Tensile stress-strain curves of different volume fractions of PAEM (16.6, 25.0, 33.3, and 41.6%); (**d**) Different volume fractions of PAEM (16.6, 25.0, 33.3, and 41.6%); (**e**) variation of conductivity for different mass fractions of PAEM (0.075, 0.15, 0.23, 0.30, and 0.37%); (**f**) sensing properties of different mass fractions of PAEM (0.075, 0.15, 0.23, 0.30, and 0.37%).

**Figure 6 polymers-15-00250-f006:**
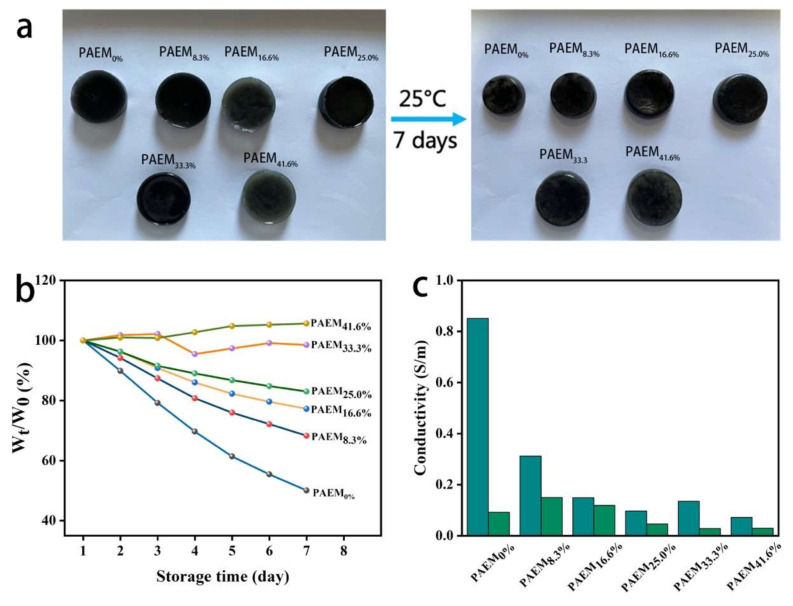
(**a**) Comparison of PAEM with different volume fractions after 7 days at 30 °C and 35% humidity; (**b**) change in conductivity of PAEM with different volume fractions and after 7 days at 30 °C and 35% humidity; (**c**) change in weight of PAEM with different volume fractions and after 7 days at 30 °C and 35% humidity. w0 is the initial weight. wt is the weight of different objects at storage time.

**Figure 7 polymers-15-00250-f007:**
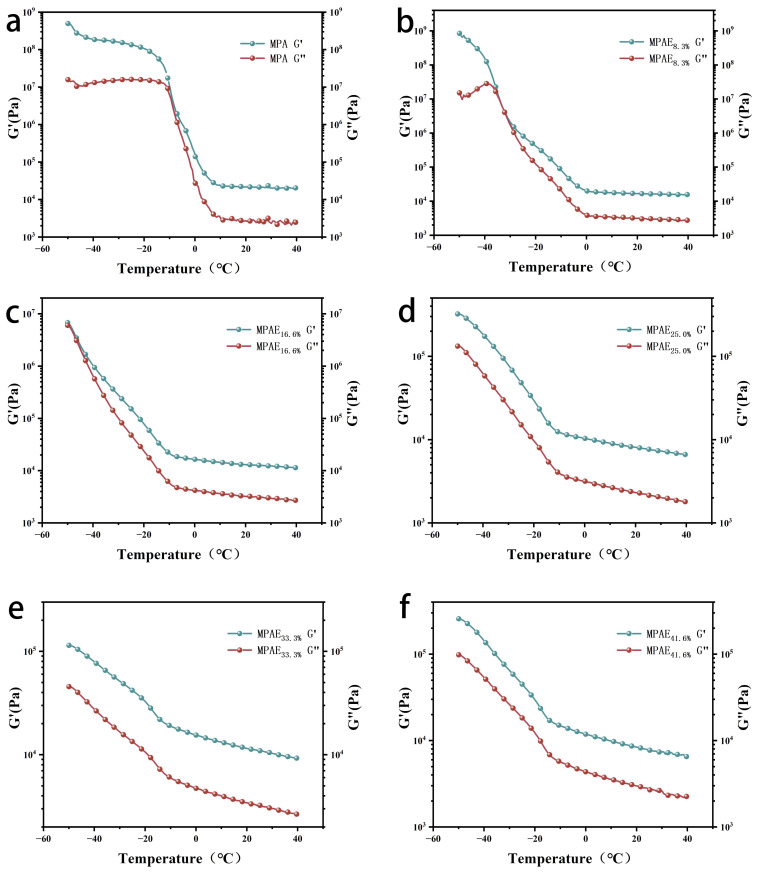
Plots of energy storage modulus (G’) and loss modulus (G″) for different volume fractions of PAEM ((**a**) 0%, (**b**) 8.3%, (**c**) 16.6%, (**d**) 25%, (**e**) 33.3%, and (**f**) 41.6%) at temperatures ranging from −50 to 40 °C.

**Figure 8 polymers-15-00250-f008:**
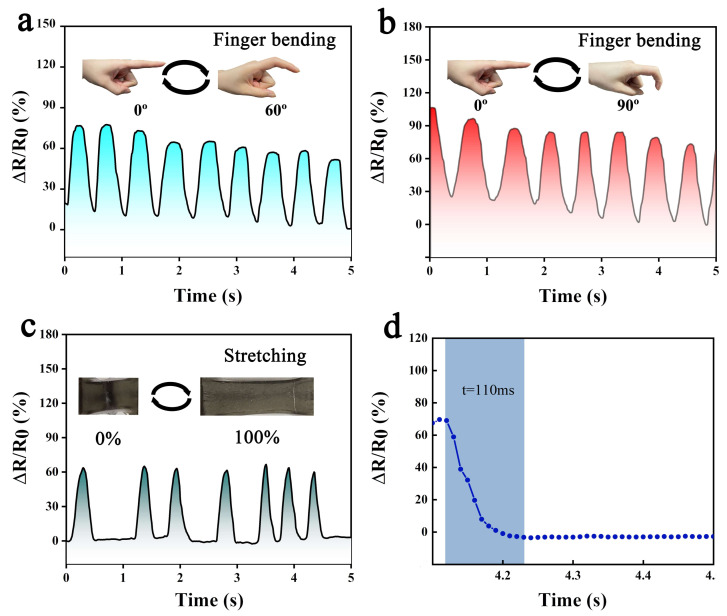
(**a**) Sensing performance at 60° finger flexion; (**b**) Sensing performance at 90° finger flexion; (**c**) Sensing performance of PAEM stretched to 100%; (**d**) Response time of PAEM hydrogel.

## Data Availability

The data presented in this study are available on request from the corresponding author. The data are not publicly available due to no new data were created.

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
