# Peer review of "Flexible Stretchable, Dry-Resistant MXene Nanocomposite Conductive Hydrogel for Human Motion Monitoring"

_polymers, 2023, doi:10.3390/polym15020250_

Round 1

Reviewer 1 Report

Liu et al. have presented the manuscript titled: Flexible stretchable, dry-resistant MXene nanocomposite conductive hydrogel for human motion monitoring. Overall presentation of the article is good, but there require few modification before being publish, suggestions are as follow;

1.       In introduction section, second sentence of first paragraph is presenting no sense, there are grammatical mistakes, please correct it.

2.      In second paragraph of introduction,” Li et al.[16] developed an organic conductive hydrogel…..”, authors should mention the reported values of electrical conductivity and mechanical property so that it will help the readers to compare the achieved results and reported results.

3.      Paragraph 2, lines 40-42, “However, the lack of conductive path in traditional hydrogels leads to low conductivity because of the short of appropriate medium such conductive polymer….”, there are no understandings for this sentence, please make it smooth and avoid grammatical and technical errors.

4.      In the method sections (manuscript, not in supplementary information) please add the specs and parameters used for the measurements of XRD, XPS, FTIR etc.

5.      Add the manufacturer, company and purity of the chemicals added for this research.

6.      Please provide the EDX analysis to confirm the constituent ratios.

7.      For XRD analysis, please provide the PDF card number for the comparison of indexing of peaks.

Author Response

Point 1: In introduction section, second sentence of first paragraph is presenting no sense, there are grammatical mistakes, please correct it.

Response 1: We think this is an excellent suggestion.We huve re-written this part according to the Reviewer’s suggestion. Lines 29–30 of the manuscript contain information.

Point 2: In second paragraph of introduction,” Li et al.[16] developed an organic conductive hydrogel…..”, authors should mention the reported values of electrical conductivity and mechanical property so that it will help the readers to compare the achieved results and reported results.

Response 2: : We think this is an excellent suggestion.We huve re-written this part according to the Reviewer’s suggestion. Lines 36–39 of the manuscript contain information.

Point 3: Paragraph 2, lines 40-42, “However, the lack of conductive path in traditional hydrogels leads to low conductivity because of the short of appropriate medium such conductive polymer….”, there are no understandings for this sentence, please make it smooth and avoid grammatical and technical errors.

Response 3: We tried our hardest to improve the manuscript and addressed the issues you raised. These modifications have no bearing on the paper's content or structure. Lines 39–42 of the manuscript contain information. We sincerely thank the reviewers for their dedicated efforts and hope that the corrections are accepted.

Point 4: In the method sections (manuscript, not in supplementary information) please add the specs and parameters used for the measurements of XRD, XPS, FTIR etc.

Response 4: We sincerely appreciate your attentive reading, and we have updated the manuscript with specifications and parameters for XRD, XPS, FTIR, etc.

Point 5: Add the manufacturer, company and purity of the chemicals added for this research.

Response 5: We appreciate your offer from the bottom of our hearts. We have included in the manuscript the manufacturer, company and purity of the added chemicals.

Point 6: Please provide the EDX analysis to confirm the constituent ratios.

Response 6: We sincerely appreciate you for your suggestions, and please see lines 217-218 in the manuscript for the ratios of each element. Because of the epidemic's impact, the experimental drugs are currently in the laboratory, so please accept our apologies for being unable to provide EDX tests. We have determined the content of each element based on XPS data analysis.

Point 7: For XRD analysis, please provide the PDF card number for the comparison of indexing of peaks.

Response 7: We appreciate your feedback and have revised the relevant content in the manuscript.

Reviewer 2 Report

This paper present the preparation and performances of MXene nanosheet reinforced polyvinyl alcohol/polyacrylamide conductive hydrogels. The composite hydrogel has good mechanical performance, temperature tolerance and electrical property. The authors should further clarify the novelty of the work considering the existence of some MXene-hydrogel composite papers. The English writing should also be improved. The paper should be acceptable to Polymer after the improvement. Below are some detail comments to the manuscript.

 1、 Line 154-159, the result of sensing sensitivity is confusing. The sensitivity dropped severely from PAEM0.30% to PAEM0.37%, which is of the same level with PAEM0.075%. It is too simple to conclude the decrease only as stacking of nanosheets and the sharp decrease within this range worth further probing.

2、 Line 208-216, the result of storage modulus and loss modulus suggested that the icing temperature of PAEM at -18 ℃. With the modulus increasing rapidly at temperature below that, it is unsuitable to describe PAEM as “stable temperature tolerance from -50 to 40 ℃”. Thus, the result in Figure S3 is susceptible. The author should demonstrate the exact temperature when tested or provide detailed time of balance in the experiments conducted using DMA.

3、 Line 220-229, if the MXene was in contact with the hydrogel network, how would it be like if the gel was stretched in different orientation? Will opposite results be acquired if the gel was stretched in an orientation vertical to the present one?

4、 Spelling mistakes like MPAE in Line 187.

Author Response

Point 1: Line 154-159, the result of sensing sensitivity is confusing. The sensitivity dropped severely from PAEM0.30% to PAEM0.37%, which is of the same level with PAEM0.075%. It is too simple to conclude the decrease only as stacking of nanosheets and the sharp decrease within this range worth further probing.

Response 1: We sincerely appreciate the valuable comments. The MXene nanosheets in the hydrogel cannot form a complete conductive network when a small amount of MXene material is added, and the sensitivity is low. The GF reaches its peak when MXene is added at 0.3% and then starts to decline as more MXene is added. However, when excessive MXenes are incorporated, the accumulation of MXene nanosheets inevitably impedes their sliding under deformation, resulting in an obscure reduction in contact resistance and ultimately leading to a faint GF.

Point 2:  Line 208-216, the result of storage modulus and loss modulus suggested that the icing temperature of PAEM at -18 ℃. With the modulus increasing rapidly at temperature below that, it is unsuitable to describe PAEM as “stable temperature tolerance from -50 to 40 ℃”. Thus, the result in Figure S3 is susceptible. The author should demonstrate the exact temperature when tested or provide detailed time of balance in the experiments conducted using DMA.

Response 2: I appreciate your review comments very much. Figure 7a shows that the modulus of PAEM hydrogel without EG increases significantly below -18°C and is at a high level (>107 Pa), indicating that the material's freezing point is at -18°C. However, it can be seen from the data in Figure 7e that PAEM maintains a lower modulus (105 Pa) as the EG content increases, showing that the strength and elasticity can be preserved at low temperatures. Figure S3 depicts our demonstration experiments in which the hydrogel underwent tensile tests after being stored at -40°C and still exhibited some elasticity. Temperature changes will somewhat impact the performance of PAEM; however, PAEM can still function to some extent between -50 and 40°C.

Point 3: Line 220-229, if the MXene was in contact with the hydrogel network, how would it be like if the gel was stretched in different orientation? Will opposite results be acquired if the gel was stretched in an orientation vertical to the present one?

Response 3: We sincerely appreciate your insightful remarks, as you have already stated. We have also conducted experiments in the stretching direction that is perpendicular to the one we are currently in, and we will obtain the opposite outcome. In our tests, the hydrogel's resistance will rise above its initial value when stretched and strained, and it will fall below its initial value when compressive strain is applied.

Point 4: Spelling mistakes like MPAE in Line 187.

Response 4: We sincerely apologize for our careless actions. We appreciate the reminder. We sincerely appreciate the reviewer's attentive reading. We changed the "MPAE" to "PAEM" as the reviewer suggested.

Reviewer 3 Report

The research article ‘Flexible stretchable, dry-resistant MXene nanocomposite conductive hydrogel for human motion monitoring’

In this research, the fabrication of MXenes-polymers conductive hydrogel is reported. The Authors aim MXenes application in the formation of hydrogel process could improve properties and extend application areas. Therefore, some characterization methods were employed for the investigation of samples. The topic of this work is novel, and the interest in the MXenes and their applications is growing at the moment.

Some remarks about this manuscript:

·       The synthesis of MXenes is done quite in an opposite way than in most of the published articles from inventors of MXenes prof. Gogotsi and his group. Why sediment after synthesis was collected instead of supernatant, where should delaminated MXenes stay? It is reported that in sediment, there are located unetched MAX phase and multi-layered MXenes. Please check: 10.1021/acs.chemmater.7b02847.

·       Comparing 10.1021/acs.chemmater.7b02847 Fig. 5. in the Gogotsi paper, do not match XRD data in this manuscript, how could these differences be explained?

·       Image 1 is misleading in this manuscript. It shows the traditional way of MXenes synthesis if, after delamination in the presence of lithium ions, the supernatant was collected. In this manuscript, the synthesis process is done differently and ultrasonication is used.

·       Characterization of MXenes part is weak; SEM/TEM images of used MXenes are needed. It is not clear what size and shape MXenes were synthesized. Ultrasonication should reduce the size MXenes sheets.

·       centrifuged at 3500 rpm” – instead of rpm, the G factor should be calculated.

·       Fig 2. A bottom part, TiAlC2 – should be changed into Ti3AlC2.

·       What was the conductivity of synthesized MXenes?

·       The MM section needs more information about equipment used for characterization and testing.

·       Information about materials used in this study should be added in the MM section.

·       Please, fix typos in the whole text.

Author Response

Point 1: The synthesis of MXenes is done quite in an opposite way than in most of the published articles from inventors of MXenes prof. Gogotsi and his group. Why sediment after synthesis was collected instead of supernatant, where should delaminated MXenes stay? It is reported that in sediment, there are located unetched MAX phase and multi-layered MXenes. Please check: 10.1021/acs.chemmater.7b02847.

Response 1: We sincerely apologize for our carelessness. Thank you for reminding me. We made a typo in the manuscript, and the MXene we used was the deposit obtained from etching, not the supernatant. Again, we apologize for our carelessness and hope you will accept our corrections.

Point 2: Comparing 10.1021/acs.chemmater.7b02847 Fig. 5. in the Gogotsi paper, do not match XRD data in this manuscript, how could these differences be explained?

Response 2: We sincerely appreciate your inquiries. We carefully read the XRD data in 10.1021/acs.chemmater.7b02847 Fig. 5 of the manuscript, and all three MXenes mentioned there are the products of TMAOH delamination. Similarly, 10.1021/acs.chemmater.7b02847 Fig. 4 in the manuscript shows that the shift of the (002) peak to a smaller angle and the weakening of the (104) Al feature peak at 39° in the acid-etched MXene sample compared to Ti3AlC2 is primarily due to the successful etching of the aluminum layer and the embedding of lithium ions, indicating the successful etching of the MXene material. This is consistent with our findings in the manuscript.

Point 3: Image 1 is misleading in this manuscript. It shows the traditional way of MXenes synthesis if, after delamination in the presence of lithium ions, the supernatant was collected. In this manuscript, the synthesis process is done differently and ultrasonication is used.

Response 3: We sincerely appreciate your feedback and apologize for any trouble caused by our mistakes. The traditional method of synthesis of MXene is used in this manuscript.

Point 4: Characterization of MXenes part is weak; SEM/TEM images of used MXenes are needed. It is not clear what size and shape MXenes were synthesized. Ultrasonication should reduce the size MXenes sheets.

Response 4: We sincerely appreciate your suggestion, and for your review, we have added the MXene electron microscopy images to the manuscript.

Point 5: “centrifuged at 3500 rpm” – instead of rpm, the G factor should be calculated.

Response 5: We sincerely appreciate your comments, and after your question, we have reviewed the papers on MXene hydrogels, where the centrifugal speed is primarily expressed in "rpm". Appreciate you your valuable suggestions.

Point 6: Fig 2. A bottom part, TiAlC2 – should be changed into Ti3AlC2.

Response 6: We apologize for the mistakes and appreciate your reminders. We sincerely appreciate the reviewers' careful reading and have updated it in the manuscript.

Point 7: What was the conductivity of synthesized MXenes?

Response 7: 2000 S/cm

Point 8: The MM section needs more information about equipment used for characterization and testing.

Response 8: We appreciate your suggestion and have updated the manuscript with characterization and test tools details.

Point 9: Information about materials used in this study should be added in the MM section.

Response 9: We sincerely appreciate your thorough review, and we have updated the manuscript with details on the pertinent materials.

Point 10: Please, fix typos in the whole text.

Response 10: We apologize for our careless errors and thank the reviewers for their thorough reading. We reorganized the manuscript once more to correct the mistakes.

Round 2

Reviewer 3 Report

The manuscript was improved and ready for publication.